# Onychomycosis in Diabetics: A Common Infection with Potentially Serious Complications

**DOI:** 10.3390/life15081285

**Published:** 2025-08-13

**Authors:** Aditya K. Gupta, Amanda Liddy, Lee Magal, Avner Shemer, Elizabeth A. Cooper, Ditte Marie L. Saunte, Tong Wang

**Affiliations:** 1Mediprobe Research Inc., London, ON N5X 2P1, Canada; aliddy@mediproberesearch.com (A.L.); lcooper@mediproberesearch.com (E.A.C.); twang@mediproberesearch.com (T.W.); 2Division of Dermatology, Department of Medicine, Temerty Faculty of Medicine, University of Toronto, Toronto, ON M5S 1A8, Canada; 3Shemer Clinic, Netanya 4216000, Israel; leemagal@gmail.com; 4Department of Dermatology, Sheba Medical Center, Tel-Hashomer, Ramat-Gan 52621, Israel; ashemer1@gmail.com; 5Sackler Faculty of Medicine, Tel Aviv University, Tel Aviv 6997801, Israel; 6Department of Clinical Medicine, Faculty of Health and Medical Sciences, University of Copenhagen, 2100 Copenhagen, Denmark; ditte.marie.saunte@regionh.dk; 7Department of Dermatology and Allergy, Copenhagen University Hospital—Herlev and Gentofte, 2900 Gentofte, Denmark

**Keywords:** diabetes, onychomycosis, treatment, antifungals, management, prophylaxis, disinfection

## Abstract

Onychomycosis is a prevalent and clinically relevant complication among individuals with diabetes. It is associated with an elevated risk of secondary fungal and bacterial infections, foot ulceration, and, in advanced cases, amputation. Factors contributing to the increased prevalence of onychomycosis in this population include age, peripheral vascular disease, poor glycemic control, neuropathy, suboptimal foot hygiene, and nail trauma. While dermatophytes are the most common pathogens, diabetic patients are more prone to mixed infections involving *Candida* species with varying antifungal susceptibility profiles, necessitating accurate identification to guide therapy. Prompt diagnosis and early intervention are important to prevent complications. Systemic antifungals such as terbinafine and itraconazole are considered first-line therapies, particularly for moderate to severe onychomycosis. However, drug interactions, renal, hepatic, and metabolic comorbidities may necessitate individualized treatment plans. For patients with mild to moderate disease, or contraindications to oral therapy, topical agents such as efinaconazole or tavaborole offer viable alternatives. Adjunctive measures, including education on foot hygiene, prompt treatment of tinea pedis, and environmental sanitization, are important in preventing recurrence and reinfection. This review summarizes the epidemiology, diagnosis, and treatment considerations for onychomycosis in diabetic patients, emphasizing the need for individualized care to improve outcomes in this high-risk population.

## 1. Introduction

Diabetes mellitus is one of the most common chronic metabolic diseases worldwide, and continues to increase in prevalence at an alarming rate [1,2]. The International Diabetes Foundation (IDF) estimated that the global prevalence of diabetes in 2024 was 589 million [3]. Additionally, it was estimated that 240 million individuals live with undiagnosed diabetes [4]. The countries with the highest prevalence of diabetes globally include the United States, India, China, Indonesia, Japan, Pakistan, Russia, Brazil, Italy, and Bangladesh [3].

In the U.S., diabetes affects an estimated one in six adults, with a reported prevalence of 15.8% between August 2021 and August 2023 [5]. It was the eighth leading cause of death in 2020, accounting for more than 100,000 deaths [6]. Diabetes contributes significantly to cardiovascular complications, microvascular damage, and premature mortality [7]. Its systemic effects compromise immune function, rendering diabetic patients more susceptible to both local and systemic infections, which tend to present more severely than in the general population [2].

In diabetics, around 30% of patients experience dermatological manifestations, with fungal and bacterial skin infections occurring more frequently and progressing more aggressively than in those without diabetes [2,8]. As diabetes continues to rise, increased attention is being given to under-recognized secondary complications like onychomycosis, which disproportionately affects individuals with lower income and limited education, and those identifying as Black or Hispanic [9]. Onychomycosis is among the most common nail disorders, impacting an estimated 4–5% of people worldwide [1,9,10]. Studies consistently show that individuals with diabetes face a substantially greater likelihood of developing onychomycosis, with some estimates placing the burden at up to three times that of people without diabetes [9,11,12,13,14]. Notably, diabetes is the leading cause of lower-limb amputation in the United States, most often following a non-healing foot ulcer [15]. Onychomycosis may contribute to these ulcers by compromising nail and skin integrity and increasing the risk of secondary infections [16,17].

Focusing on diabetic individuals, this review explores the burden, diagnostic challenges, and therapeutic approaches to onychomycosis, highlighting the importance of tailored management strategies to enhance patient outcomes in this vulnerable group.

## 2. Epidemiology

### 2.1. Prevalence of Onychomycosis in Diabetic Patients

The prevalence of onychomycosis is notably higher in individuals with diabetes compared to the general population, with several studies highlighting demographic trends and risk patterns within this population. In a cohort study of 550 diabetic patients, Gupta et al. observed toenail abnormalities in 46% of participants and confirmed onychomycosis in 26% compared to 9.1% of confirmed diagnoses in non-diabetic individuals [11]. Among those with type 1 diabetes—comprising 34% of the cohort—the prevalence was lower, at 13%, suggesting that disease type, duration, and metabolic profile may influence susceptibility and severity [11]. Additionally, Agrawal et al. reported a significantly higher prevalence of onychomycosis among male diabetic patients, accounting for 61.8% of cases compared to 38.2% in females [1]. They also observed that patients with onychomycosis were, on average, older than those without onychomycosis, indicating an age-related increase in prevalence coinciding with the age-related increase in diabetes incidence [1]. These findings echo data from a number of independent studies [11,18,19,20].

### 2.2. Diagnostic Approaches and Fungal Identification

Diagnosing onychomycosis in diabetic patients does not differ from that of the general population. Currently employed diagnostic methods, including KOH microscopy, histopathology, fungal culture, and molecular methods such as polymerase chain reaction (PCR), have inherent limitations with regards to cost, the level of expertise required to perform the diagnostic testing, the amount of time it takes to obtain test results, or a combination of the three. While fungal culture remains the gold standard for species-level identification and confirmation of viability, molecular methods, including PCR targeting the Internal Transcribed Spacer (ITS) region, offer enhanced sensitivity and the ability to detect a broad range of pathogens, including dermatophytes, yeasts, and non-dermatophyte molds (NDMs). [21,22,23].

In the general population, dermatophytes, particularly *Trichophyton rubrum* and *T. mentagrophytes*, are the leading causes of onychomycosis, followed by less common organisms such as *Epidermophyton floccosum*, NDMs like *Aspergillus*, *Fusarium*, and *Scopulariopsis brevicaulis*, and yeasts such as *Candida* spp., which are more frequently implicated in fingernail infections [24,25]. Mirroring trends in the general population, dermatophytes are the most common causative agents of onychomycosis in diabetics [13,15,26]. However, studies have demonstrated that diabetic patients may exhibit greater susceptibility to mixed infections, particularly those involving *Candida* spp. [13,27,28], along with uncommon pathogens such as *Kodamaea ohmeri* and *Prototheca wickerhamii* [29]. Distal lateral subungual onychomycosis (DLSO) remains the most frequently observed morphological subtype [1,19,28,30,31].

Several reports and cohort studies have highlighted the prevalence and significance of non-dermatophyte mold (NDM) onychomycosis in patients with diabetes. Wu et al. described a severe case of disseminated *Fusarium solani* onychomycosis in a diabetic woman, which progressed to subcutaneous tissue involvement with bacterial superinfection, ultimately leading to osteomyelitis and septic shock [32]. In a cohort study involving 300 consecutive diabetic patients attending a dermatology clinic in India, 30 of 63 isolated organisms were NDMs (46.7%) [28]. The identified organisms included *Aspergillus (A.) flavus*, *A. niger*, *A. versicolor*, *Trichosporon*, *Chaetomium*, *Cladosporium*, *Geotrichum*, *Fonsecaea*, *Fusarium solani* complex, and *Scopulariopsis brevicaulis* [28].

### 2.3. Pathophysiology and Clinical Presentation of Onychomycosis in Diabetics

Diabetic patients are at increased risk of onychomycosis, due to a combination of systemic, vascular, and behavioral factors. Established risk factors include older age, sex, peripheral vascular disease, obesity, and poor glycemic control [1,15]. Impaired circulation, reduced peripheral sensation, and immune dysfunction further predispose diabetic patients to fungal infections [26]. Additionally, poor foot hygiene, trauma, and inadequate adherence to foot-care practices exacerbate this risk, particularly in patients with sensory neuropathy, who may not notice early signs of nail or skin trauma. Early diagnosis, appropriate glycemic management, and timely antifungal therapy can significantly reduce the morbidity associated with fungal infections [2].

Onychomycosis and tinea pedis frequently coexist and share common causative dermatophytes, most notably *Trichophyton rubrum*. The presence of tinea pedis is a well-established risk factor for developing onychomycosis, as fungal elements can spread from the skin of the foot to the nail unit [33,34]. Conversely, untreated onychomycosis can act as a reservoir for recurrent or persistent tinea pedis [33,34]. In diabetic patients, co-infection with both conditions further increases the risk of secondary bacterial infections and foot complications, underscoring the need for comprehensive foot examinations and simultaneous treatment strategies.

Hyperglycemia induces microvascular ischemia and impairs neutrophil and phagocyte function, contributing to susceptibility to both fungal infections and diabetic foot ulcers [2,35]. As such, maintaining glycemic control may be vital for minimizing fungal burden and preventing progression to more serious diabetic foot complications. A recent study reported a statistically significant correlation between elevated HbA1c levels and increased number of nails involved in the infection (*p* = 0.014, r = 0.14) [1].

At the molecular level, one hypothesis is that persistent hyperglycemia leads to the formation and accumulation of advanced glycation end-products (AGEs) within the extracellular matrix [36]. Elevated AGEs may impair local immune responses by disrupting antigen presentation and dendritic cell recognition of fungal pathogens, and, when combined with the immune-privileged nature of the nail unit, could increase the susceptibility to fungal infections [36,37,38]. Through binding to nail proteins such as keratins, Cosio et al. hypothesized that AGEs also potentiate fungal adhesion by providing binding targets—such as mannose and galactose residues—for adhesins expressed on conidia [37,39,40]. Glycated nail proteins are shown to be increased in diabetic patients compared to non-diabetic patients, and have been proposed as a diagnostic and prognostic factor [39,41]. Even for patients with controlled diabetes, high levels of nail glycation due to AGEs may persist, highlighting the importance of early glycemic control [39,42].

### 2.4. Onychomycosis Complications in Diabetics

Though often considered minor in the general population, superficial fungal infections like tinea pedis and onychomycosis carry significantly greater risks in diabetic patients. Thickened, dystrophic nails can inflict mechanical trauma on adjacent skin and contribute to pressure ulcers by increasing localized stress on an already compromised vascular bed, especially in patients with reduced foot sensation [13,15,43]. Chronic erosions associated with these infections impair wound healing and create conditions favorable for polymicrobial colonization, potentially leading to serious infections [13,15,31].

Foot ulcers are among the most debilitating sequelae of diabetes, representing a major contributor to hospitalization and disability [44]. Up to 50% of individuals with diabetes develop peripheral neuropathy, approximately one-third will experience a foot ulcer, and of those ulcers, nearly half become infected, with as many as 20% resulting in amputation [44,45,46,47]. Tinea pedis and onychomycosis are independent predictors of foot ulceration [46,48], and their presence may accelerate a clinical cascade that culminates in tissue breakdown (Figure 1). Navarro-Pérez et al. reported that diabetic patients with onychomycosis had significantly higher odds of a history of minor amputation (OR = 4.493, *p* = 0.014; 95% CI: 1.356–14.881), underscoring the clinical relevance of timely diagnosis and intervention [13]. In the U.S. Medicare population, lower extremity amputations (LEAs) were associated with significantly elevated annual mortality rates, at 170 per 1000 for patients with prevalent LEAs, and 206 per 1000 for incident LEA cases [49]. Disparities in LEA rates among racial and ethnic minorities, as well as the high post-amputation mortality rate, further highlight the need for proactive management of fungal nail infections in at-risk diabetic populations.

## 3. Treatment of Onychomycosis in Diabetic Patients

### 3.1. Tailoring Therapy for Diabetics

While the causative organisms of onychomycosis in diabetic individuals are largely similar to those in the general population, management strategies must be adapted to reflect the unique clinical challenges posed by diabetes [15,26]. Chronic hyperglycemia promotes fungal growth, impairs immune response, and diminishes treatment efficacy, thereby complicating clinical outcomes and often prolonging therapeutic timelines [16].

Treatment in diabetics requires a comprehensive, individualized approach that extends beyond pharmacologic intervention. There needs to be foot-care optimization, patient education on proper hygiene, and early recognition of complications [50]. Shared decision-making that incorporates patient preferences and understanding of the risks and benefits of treatment is essential in optimizing outcomes for this high-risk population.

### 3.2. Oral Antifungals

Systemic antifungal therapy in diabetic patients requires careful consideration of both efficacy and safety, particularly given the potential for drug–disease and drug–drug interactions (Table 1). Among systemic agents, terbinafine is frequently considered a first-line option in diabetic populations, due to its favorable efficacy and safety profile. Terbinafine is generally prescribed at a dose of 250 mg once daily for 6 weeks to treat fingernail onychomycosis and for 12 weeks to treat toenail onychomycosis [51]. Farkas et al. reported that in a cohort of diabetic patients treated with terbinafine 250 mg daily for 12 weeks there were no hypoglycemic episodes and no drug interactions despite the use of multiple concomitant medications in the study participants [52]. Additionally, within this cohort of patients, the reported mycological cure rate was 73%, further underscoring the efficacy of terbinafine in addition to minimal impact on glycemic interactions and low potential for pharmacologic interactions [52]. A review of the existing evidence on terbinafine in high-risk groups, including patients with diabetes, demonstrated similar mycological cure rates in diabetic versus non-diabetic individuals (64% vs. 73%), suggesting that diabetes does not markedly diminish treatment effectiveness [53]. A comparable pattern was noted for clinical cure rates as well (37% vs. 45%, respectively).

While terbinafine is generally safe and effective in diabetic patients, it poses potential risks due to its inhibition of CYP2D6, which may impair the metabolism of several drugs. While no specific drugs are absolutely contraindicated, co-administration with beta blockers, immunosuppressants, antitussives, antipsychotics, anxiolytics, and certain antifungals may lead to elevated plasma concentrations of the co-administered drugs [51]. This is particularly concerning for agents with a narrow therapeutic index, such as select antidepressants and antiarrhythmics, where increased levels may heighten the risk of adverse effects [51]. As such, it is suggested that patients taking these drugs concomitantly with terbinafine be monitored for adverse effects and that dosage reductions of the concomitant drug be performed, if necessary.

Itraconazole, like terbinafine, has been effectively utilized in diabetic patients; however, its use may require more caution in view of potential drug–drug interactions. In a comparative study, Matricciani et al. demonstrated that pulse itraconazole was as effective as continuous terbinafine, without evidence of increased adverse outcomes [46]. For the treatment of onychomycosis, fingernail infections are typically managed with continuous itraconazole therapy for 6 weeks, or with pulse itraconazole therapy (200 mg twice daily for 1 week, followed by a 3-week drug-free interval) over a total duration of 12 weeks (2 pulses) [54]. Toenail infections require continuous dosing of 200 mg daily for 12 weeks or pulse therapy (200 mg twice daily for 1 week, followed by a 3-week drug-free interval) for 3 pulses over a total duration of weeks [54]. Itraconazole has been used in diabetic patients with generally favorable outcomes, and most clinical data indicate a low risk of hypoglycemia during treatment [15]. Many first-line antidiabetic agents, such as metformin, gliclazide, and glipizide, are not metabolized by the CYP3A4 enzyme system, reducing the potential for pharmacokinetic interactions with itraconazole [15,55]. However, rare instances of hypoglycemic episodes have been reported, particularly in patients concurrently treated with insulin or sulfonylureas, as noted in one post-marketing surveillance study [56]. Additionally, concomitant use with repaglinide or saxagliptin should be monitored for adverse reactions and dosing adjustments may be necessary [54]. Itraconazole is contraindicated in patients receiving statins that are metabolized by CYP3A4, including simvastatin and lovastatin, due to the risk of elevated statin levels and associated hepatotoxicity, nephrotoxicity, and myopathy [54,57]. Dybro et al. described a case involving a 47-year-old woman with diabetes who developed rhabdomyolysis after a 10-day course of itraconazole (100 mg daily) while on long-term simvastatin therapy (80 mg daily) [58]. She presented with markedly elevated creatine kinase (30,855 U/L) and myoglobin (5046 mg/L; reference < 75 mg/L). Simvastatin was discontinued and she was managed with aggressive intravenous crystalloid therapy [58].

Fluconazole is not approved by the FDA for the treatment of onychomycosis, and is administered off-label at a dose of 150–200 mg once weekly for 6 to 9 months for fingernail onychomycosis and 12 to 18 months for toenail involvement [59]. In contrast to other antifungals, fluconazole has a more established potential for affecting glucose homeostasis. It inhibits the hepatic metabolism of sulfonylureas including tolbutamide, glyburide, and glipizide, potentially leading to hypoglycemia [60]. For this reason, patients with diabetes receiving fluconazole should be monitored closely for changes in blood glucose levels [26,43].

Additional booster doses of systemic antifungals may be warranted after an initial course, to improve outcomes [61,62]. A booster dose, consisting of an extra 4 weeks of terbinafine or itraconazole, may be administered 6 to 9 months after starting treatment to enhance long-term efficacy [61,63].

The presence of comorbidities such as liver or kidney dysfunction significantly complicates oral antifungal therapy in diabetic patients. Chronic kidney disease, one of the most serious and prevalent complications of diabetes, affects approximately one-third of individuals with type 1 diabetes and up to half of those with type 2 diabetes [64,65]. In patients with chronic kidney disease, renally excreted agents like fluconazole may require careful dose adjustment to avoid accumulation and toxicity, while terbinafine is generally not recommended when creatinine clearance falls below 50 mL/min due to reduced drug clearance [50,66,67]. In contrast, itraconazole has a minimal renal excretion and may be considered for diabetic patients with renal impairment [55].

In patients with pre-existing liver disease, itraconazole is not recommended for use [50]; terbinafine is contraindicated in patients with chronic or active liver disease, and fluconazole should be used with caution in patients with liver disease. In such instances, regular liver function monitoring is recommended.

### 3.3. Topical Antifungals

Topical therapies such as efinaconazole 10%, tavaborole 5%, amorolfine 5% and ciclopirox 8% are indicated for mild-to-moderate onychomycosis cases, especially for patients with limited nail area involvement [62,68]. Given the increased risks of complications in diabetic patients, particularly those with peripheral vasculopathy, neuropathy or poor glycemic control, the use of systemic vs. topical agents needs to be carefully evaluated. Topical agents are not generally recommended for severe or extensive onychomycosis, but they remain a valuable and safe alternative to oral antifungals [69,70]. 

Topical antifungals are formulated for localized treatment and exhibit negligible systemic absorption, thereby minimizing the potential for pharmacokinetic interactions with concurrent medication [26,69,70]. This makes them especially advantageous in individuals with hepatic or renal compromise or in those managing multiple comorbidities requiring complex medication regimens [50,71]. They need to be applied daily for 12 months [62,69,70]. However, their use may be problematic for patients who are obese or have diabetic retinopathy, as these conditions can hinder the patient’s ability to inspect their feet and properly apply medication to affected nails [26,43,71].

Among available topical agents, ciclopirox 8% was the first to gain FDA approval for onychomycosis not involving the lunula and has demonstrated the ability to penetrate all layers of the nail plate with repeated use [72,73]. Topical treatment with ciclopirox 8% lacquer applied daily over twelve months led to a substantial reduction in nail involvement among diabetic patients, with the surface area affected decreasing from 64.3% to 25.7% in a cohort of 215 individuals [74]. Amorolfine is approved in many countries worldwide, and is available both over the counter and by prescription for the treatment of onychomycosis; however, it is not FDA-approved [24]. It demonstrates activity against dermatophytes, certain yeasts, and molds, with reported complete cure rates of approximately 40% [24,75,76]. In a European study examining treatment adherence among patients with onychomycosis, Schaller et al. found that significantly more patients followed the recommended use of amorolfine (85%) compared to ciclopirox (60%) [68].

Efinaconazole 10% solution, which is approved for use in the US, Canada, Japan, South Korea and Hong Kong has demonstrated superior antifungal efficacy when compared to ciclopirox and tavaborole (approved for use in US) [77,78]. An in vitro study comparing efinaconazole, ciclopirox, and amorolfine for onychomycosis found that efinaconazole had significantly lower keratin binding, with an unbound drug fraction over six times higher and a faster release rate from keratin compared to ciclopirox and amorolfine [79]. This lower keratin affinity correlated with enhanced nail penetration and greater fungicidal activity in keratin-rich environments, supporting its clinical effectiveness [79]. In a prospective study involving 40 individuals with diabetes, over half achieved mycologic clearance following 40 weeks of treatment, with subsequent follow-up at week 50 revealing clinical improvement or complete resolution in a proportion of cases [80]. Additionally, a post-hoc analysis of a randomized, double-blind, vehicle-controlled trial confirmed the superior therapeutic performance of efinaconazole compared to placebo, with minimal treatment-related adverse effects observed in the diabetic subgroup [16]. 

Diabetic patients ºare likely to require extended onychomycosis therapy versus non-diabetics, particularly when topical antifungals are the primary therapy. Efinaconazole has demonstrated favorable efficacy and safety when used daily beyond 12 months, and when used as prophylaxis for up to 24 months. Gupta and Cooper reported that efinaconazole 10% solution demonstrated increased antifungal efficacy over 24 months of daily use [81]. The treatment was well-tolerated, with only localized adverse events, and no systemic side effects or drug interactions [81]. In a follow-up study of 48 months, clinical cure was maintained in six patients who had achieved it by Month 24 with intermittent maintenance therapy, and three additional patients reached clinical cure by Month 48, without any increase in adverse events, with prolonged use [82].

### 3.4. Combination Treatment

In cases of extensive nail involvement, infection of multiple digits, or spread to surrounding areas, a more aggressive therapy approach combining systemic and topical antifungal agents may be a consideration [50]. Topical agents can also play a valuable adjunctive role in these complex cases—not only to enhance primary treatment efficacy when paired with oral antifungals with a different mode of action, but also as maintenance therapy aimed at reducing the likelihood of relapse [81,83].

In a retrospective case-series patient treated with oral terbinafine (the standard Japanese dosing of 125 mg daily for ≥20 weeks) followed by topical efinaconazole, results showed superior improvements in nail clearance and cure rates compared to those receiving oral therapy alone [84]. This strategy is particularly relevant in the context of non-dermatophyte mold infections, which exhibit varying degrees of resistance towards terbinafine and azoles [85,86]. 

Surgical nail avulsion should generally be avoided in diabetic patients, due to the elevated risk of complications such as secondary infections and ingrown toenails, which may be exacerbated by impaired would healing [26,87]. Conservative management with pharmacologic therapies and non-invasive debridement (such as chemical nail avulsion with 40% urea) is preferred, reserving surgical intervention only for select cases where medical treatment has failed and when the benefits clearly outweigh the risks, considering the severity of the disease and individual patient comorbidities [26,71]. 

Emerging physical modalities, including laser and photodynamic therapy, have been explored as adjunctive options for onychomycosis management. While some laser devices are approved in the United States to enhance the cosmetic appearance of the nail, they have not been proven to provide a significant or lasting antifungal effect. Importantly, laser therapy is typically contraindicated in diabetic patients because of the increased risk of thermal injury, which may precipitate ulceration and further complications [69]. 

## 4. Improving Outcomes in Diabetic Patients

### 4.1. Early Diagnosis

Traumatic and onychomycotic lesions may frequently go unrecognized in diabetic patients, particularly those with peripheral neuropathy, as impaired sensation can mask damage to the skin and nails. The characteristic thickening and brittleness of infected nails can further contribute to unnoticed skin trauma, creating portals of entry for bacterial and fungal pathogens that may lead to systemic infection [15,17]. Given the increased susceptibility of immunocompromised individuals, including diabetic patients, to severe infections and associated complications, potentially culminating in mortality, it is critical to maintain a high index of suspicion for fungal infections during cutaneous examination [17].

Considering the elevated risk of lower-limb complications linked to fungal infections such as tinea pedis and onychomycosis in this population, timely detection through routine foot and nail assessments combined with appropriate diagnostic investigations is important [46]. Prompt initiation of treatment is required when fungal infection is suspected, to reduce the risk of progression and prevent serious outcomes (Figure 2) [15].

### 4.2. Preventative Strategies and Preventing Relapse

Effective long-term management of dermatophyte infections in individuals with diabetes hinges on the prompt identification and treatment of coexisting conditions such as tinea pedis, which often serve as fungal reservoirs (Figure 3). If left untreated, these reservoirs significantly increase the risk of persistent or recurrent onychomycosis, complicating clinical outcomes and prolonging patient morbidity [88].

Although diabetics make more healthcare visits on average than the general population (15.5 vs. 5.5 per year), foot-related issues are often overlooked, due to competing demands for attention from other comorbidities [89]. Raising awareness of the complications associated with fungal infections may encourage earlier diagnosis, treatment, or referral, which not only helps prevent severe outcomes such as cellulitis and amputation, but also reduces fungal burden and supports more optimal treatment responses—highlighting the importance of comprehensive initial assessment and timely intervention [15,88].

Equally critical to therapeutic success is ensuring patient adherence to treatment regimens, which remains a modifiable yet frequently overlooked determinant in improving treatment outcomes and preventing recurrence. Establishing realistic expectations before initiating therapy, coupled with ongoing patient education throughout follow-up visits, has been shown to improve adherence rates [88]. Patients who are well-informed about the chronic and often recurrent nature of onychomycosis tend to be more compliant with prescribed treatments, adopt preventive behaviors, and seek timely medical attention upon symptom recurrence [88]. This education process fosters a collaborative approach to management that can ultimately enhance therapeutic success, reduce relapse rates, and enhance quality of life.

Consistent and rigorous foot hygiene is another cornerstone of effective prevention and management strategies. Daily foot washing, in particular, has been demonstrated to significantly reduce the likelihood of onychomycosis in diabetic populations, with one study reporting an odds ratio of 3.45 favoring this protective behavior [47]. However, the burden of dermatophyte infections within this group is often exacerbated by disparities in foot care education and inconsistent adherence to hygiene recommendations [2]. Older adults and men with diabetes are generally less likely to maintain consistent foot hygiene or seek timely treatment, leading to delayed diagnosis and increased complications [90]. In contrast, women with diabetes are more likely to ad-here to preventive practices, resulting in better outcomes [90]. These differences underscore the need for tailored educational strategies that reflect the behavioral, cultural, and social contexts of diverse patient populations.

Various approaches have been explored to improve foot hygiene and reduce the risk of household transmission of onychomycosis. Wearing footwear in communal areas, keeping feet dry, maintaining nail hygiene, and ensuring household contacts are examined and treated for fungal infections can help reduce transmission [91]. In diabetes care, it is essential to understand each patient’s individual beliefs and perspectives in order to deliver patient education that is personalized, relevant, and aligned with a patient-centered approach [90].

Environmental decontamination is a critical component of preventing reinfection, as fungal organisms can persist on textiles, footwear, and commonly touched household surfaces, creating reservoirs that facilitate ongoing transmission through indirect contact if not thoroughly disinfected [92,93,94]. Hammer et al. found that *T. rubrum* could still be recovered following laundering at 30 °C for 10 min, with up to 16% of the fungal burden detected in rinse water [95], whereas complete elimination of *T. mentagrophytes* complex and *T. rubrum* from textiles was achieved by laundering at 60 °C for 30 min, regardless of the detergent used [95,96,97]. *Aspergillus* species require higher temperatures, near 90 °C, though such heat may damage fabrics [96]. Taken together, these findings underscore the risk of persistent household contamination and the potential for continued transmission if laundering practices are inadequate. Additionally, the mechanical action during laundering also plays a vital role in removing fungal spores [98]. Fabric type affects fungal survival, with polyester and thick materials like towels posing greater challenges for eradication [99].

Chemical disinfectants vary in effectiveness: *T. mentagrophytes* complex shows resistance to agents like 0.5% benzalkonium chloride and cetrimide, requiring extended exposure for inactivation [100]. A study by Skaastrup et al. reported that soaking contaminated textiles in a QAC-based detergent for 24 hours resulted in complete eradication (13/13) of all viable Trichophyton species, regardless of their resistance profiles [94]. In contrast, shorter soaking durations were less effective, with disinfection rates of 46.2% (6/13) after 30 minutes and 84.6% (11/13) after 2 hours, highlighting the importance of prolonged expo-sure for optimal decontamination [94].

## 5. Future Directions

Onychomycosis and other dermatophyte infections continue to present a significant clinical challenge in individuals with diabetes, with a notably higher burden observed among certain racial and ethnic groups and populations facing socioeconomic disadvantages. Compared to White individuals, racial minorities such as Black and Hispanic individuals have an approximate 20–30% higher risk of contracting onychomycosis, which is associated with education and income level, as well as higher risks of diabetes [101,102] These disparities reflect complex, multifactorial influences rooted in broader social determinants of health, including variability in access to preventive healthcare services.

Advances in rapid molecular diagnostics and non-invasive imaging techniques hold promise for earlier and more accurate detection of onychomycosis and related fungal infections in diabetic patients, potentially facilitating more precise and timely interventions. These technologies can help clinicians distinguish fungal nail disease from other nail disorders common in diabetes, enabling earlier treatment and potentially reducing complications.

The persistent nature of onychomycosis in diabetic patients, coupled with the growing concern for antifungal resistance, highlights the critical role of effective antifungal stewardship and broader implementation of antifungal susceptibility testing (AFST). Antifungal-resistant dermatophyte strains, particularly *T. rubrum* and *T. mentagrophytes* complex, have emerged as global health concerns, with terbinafine resistance reported globally [103,104,105]. Despite its clinical value, AFST is not routinely performed, limiting the ability to tailor treatment in cases of failure [103,105]. Upon detection of terbinafine-resistant strains, treatment can be modified by increasing terbinafine dosage and du-ration or substituting with an azole antifungal [105]. Future investigations should prioritize the incorporation of antifungal susceptibility testing (AFST) into the clinical management of onychomycosis in diabetic patients, alongside the development of rapid diagnostic methods and robust resistance surveillance systems [103,104,105]. These advancements may support more tailored therapeutic approaches and enhance long-term outcomes in this high-risk population.

As antifungal resistance continues to rise globally, there is an urgent need for the development of novel antifungal agents and innovative formulations that can address the limitations of current therapies. Emerging alternatives include essential oils, nanotechnology-based delivery systems, and combination antimicrobial strategies [91,106]. Essential oils, in particular, show promise for use in diabetic patients, as they are not associated with hepatotoxicity or drug–drug interactions [107]. However, further clin-ical investigation is needed to establish their efficacy and safety. In parallel, advancements in antimicrobial textiles are gaining traction as a means to disrupt the transmission cycle of dermatophyte infections [108]. Functionalized fabrics, such as those impregnated with copper or silver, have demonstrated antifungal activity, offering an adjunctive approach to reduce reinfection risk and enhance hygiene practices [91,108,109].

A coordinated, multidisciplinary approach involving dermatologists, pharmacists, and microbiologists is essential to optimize antifungal selection, avoid unnecessary or prolonged therapy, and ensure timely treatment de-escalation [104,110,111]. This is particularly important in diabetic populations who are at increased risk of complications; well-implemented stewardship programs can improve clinical outcomes, reduce recurrence rates, and enhance institutional surveillance of resistance patterns [110,111,112].

Emerging pharmacotherapies which aid metabolic control of diabetes and its comorbidities, notably semaglutides, may confer additional benefits for foot health through improvements in factors supporting skin and nail integrity. Diabetic patients receiving semaglutides exhibited significantly lower incidences of wound-healing complications, chronic non-healing foot ulcers, and amputations at both one- and five-year follow-ups [113].

## 6. Conclusions

Onychomycosis represents a clinically significant complication in patients with diabetes, characterized by higher prevalence, increased severity, and an elevated risk of downstream complications such as foot ulcers, secondary bacterial infections, and lower-limb amputation. This increased vulnerability is driven by a combination of impaired immune function, peripheral neuropathy, vascular insufficiency, and the presence of advanced glycation end-products that disrupt antifungal defenses and promote fungal adhesion. Moreover, inadequate foot-care education, reduced hygiene adherence, and disparities in access to podiatric services further contribute to disease burden—particularly in older adults, men, and socioeconomically disadvantaged populations.

Diabetic individuals must be approached with a high index of suspicion for onychomycosis, in combination with proactive diagnosis activities, leading to management plans tailored to each patient’s comorbidities, treatment tolerability, and risk profile. While systemic antifungals such as terbinafine and itraconazole demonstrate strong efficacy, comorbid liver or kidney disease, polypharmacy, and the risk of drug–drug interactions may limit their use. In such cases, topical agents may offer a safe and effective alternative. Combination therapy and maintenance regimens may improve cure rates and reduce recurrence in refractory or severe cases. Attention to environmental decontamination—including textile hygiene and disinfection practices—is essential for preventing reinfection and curbing transmission.

Ultimately, effective onychomycosis management in diabetes requires a multidisciplinary, patient-centered strategy that includes routine foot examination, early intervention, patient education on hygiene and footwear, and equitable access to care. Proactive treatment not only improves clinical outcomes, but plays a critical role in reducing the progression to severe diabetic foot complications.

## Figures and Tables

**Figure 1 life-15-01285-f001:**
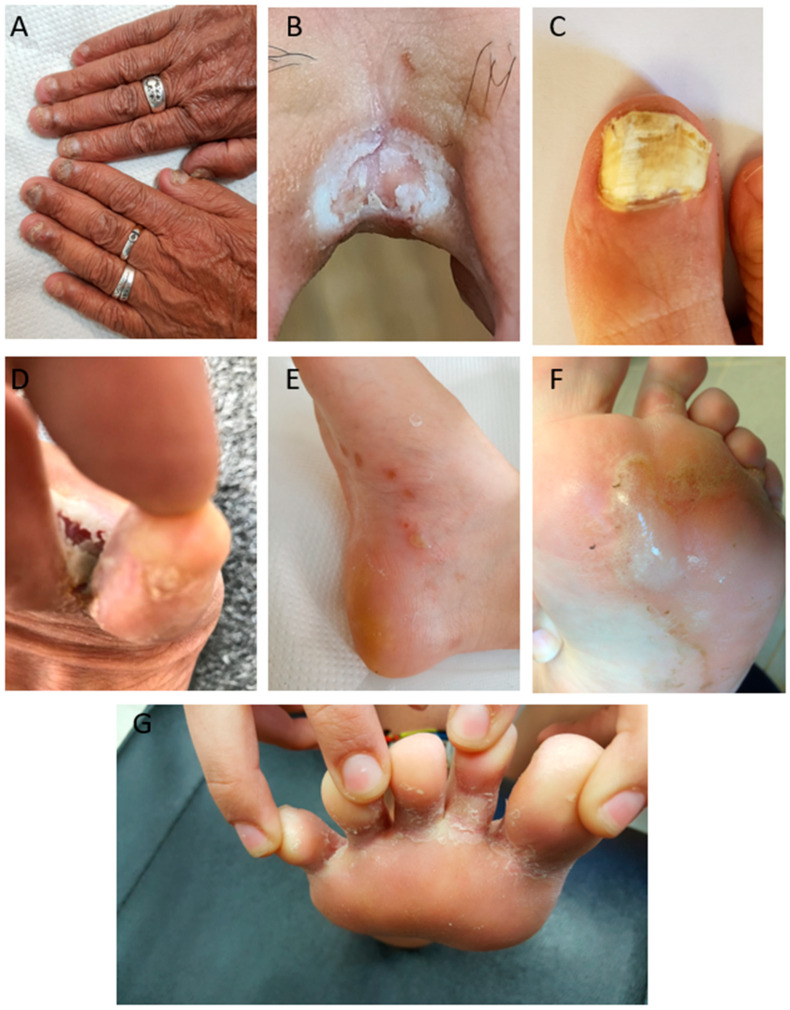
Clinical presentation of onychomycosis and tinea pedis in diabetic patients. (**A**) Fingernail onychomycosis. (**B**) Tinea pedis interdigitalis. (**C**) Severe toenail onychomycosis. (**D**) Tinea pedis ulcerative. (**E**) tinea pedis vesiculobullous. (**F**) Tinea pedis vesiculobullous. (**G**) Tinea pedis interdigitalis.

**Figure 2 life-15-01285-f002:**
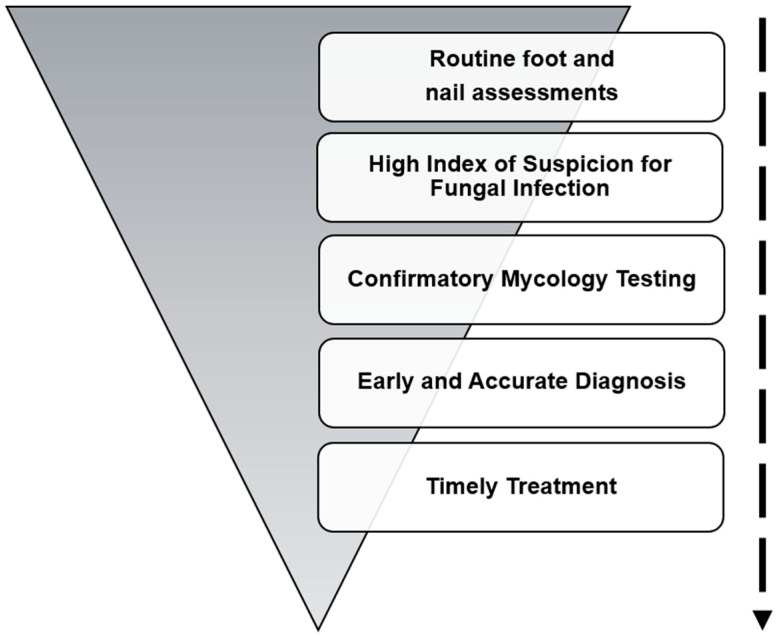
Strategies to prevent onychomycosis in diabetic patients.

**Figure 3 life-15-01285-f003:**
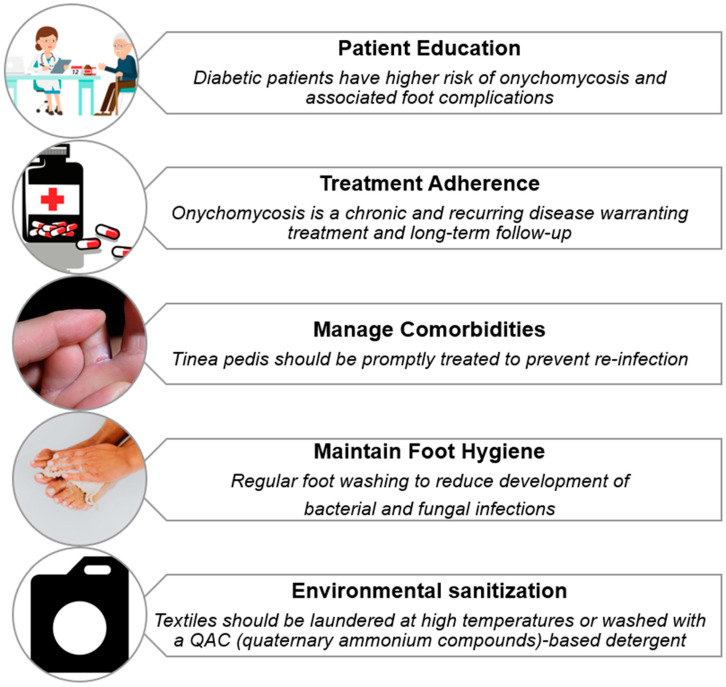
Strategies for preventing relapse of onychomycosis in diabetic patients.

**Table 1 life-15-01285-t001:** Potential drug interactions for diabetic patients taking oral antifungals*.

Antifungal	Interacting Drugs	Clinical Consideration	Interaction Mechanism
Terbinafine	Antiarrhythmics: flecainide, propafenone, amiodarone	Monitor for adverse reactions; dose reduction may be necessary	Terbinafine is a potent inhibitor of CYP2D6; co-administration with CYP2D6 substrates may reduce their metabolism, leading to elevated plasma concentrations and increased risk of adverse effects, particularly for drugs with a narrow therapeutic index such as certain antidepressants and antiarrhythmics.
	Antibacterials: rifampin		
	Antihelminthics, Antifungals and Antiprotozoals: fluconazole, ketoconazole		
	Antipsychotics, Anxiolytics and Hypnotics: drugs with CYP2D6 metabolism such as risperidone, aripiprazole, haloperidol		
	Antitussives: dextromethorphan/dextrorphan		
	Cardiovascular Drugs and Miscellaneous: beta blockers (propranolol, metoprolol, carvedilol)		
	Gastrointestinal Drugs: cimetidine		
	Immunosuppressants: cyclosporine		
	SSRIs, Tricyclics and Related Antidepressants: duloxetine, fluoxetine, paroxetine, venlafaxine, nortriptyline, desipramine, imipramine, monoamine oxidase inhibitors- Type B		
Itraconazole	Alpha Blockers: alfuzosin, silodosin, tamsulosin	Drugs metabolized through CYP3A4: contraindicated/not recommended during, and 2 weeks after, treatment	Itraconazole strongly inhibits CYP3A4; co-administration with CYP3A4 substrates may significantly reduce their metabolism, increasing systemic exposure and the potential for serious adverse effects such as QT prolongation, hepatotoxicity, or enhanced pharmacologic activity.
	Analgesics: methadone, fentanyl		
	Antiarrhythmics: disopyramide, dofetilide, dronedarone, quinidine		
	Antibacterials: bedaquiline, rifabutin		
	Anticoagulants and Antiplatelets: ticagrelor, apixaban, rivaroxaban, vorapaxar		
	Anticonvulsants: carbamazepine		
	Antihelminthics, Antifungals and Antiprotozoals: isavuconazonium, praziquantel		
	Antimigraine Drugs: Ergot alkaloids (e.g., dihydroergotamine, ergotamine)		
	Antineoplastics: irinotecan, venetoclax, mobocertinib, axitinib, ibrutinib, bosutinib, lapatinib, cabazitaxel, nilotinib, cabozantinib, olaparib, ceritinib, pazopanib, cobimetinib, sunitinib, crizotinib, trabectedin, dabrafenib, trastuzumab-emtansine, dasatinib, docetaxel, vinca alkaloids		
	Antipsychotics, Anxiolytics and Hypnotics: lurasidone, midazolam (oral), pimozide, triazolam		
	Antivirals: Elbasvir/grazoprevir		
	Calcium Channel Blockers: felodipine, nisoldipine		
	Cardiovascular Drugs and Miscellaneous: ivabradine, ranolazine, aliskiren, riociguat, sildenafil (for pulmonary hypertension), tadalafil (for pulmonary hypertension)		
	Diuretics: eplerenone, finerenone		
	Gastrointestinal Drugs: domperidone, naloxegol		
	Immunosuppressants: voclosporin, everolimus, sirolimus, temsirolimus (IV)		
	Lipid-Lowering Drugs: simvastatin, lovastatin, lomitapide		
	Miscellaneous Drugs and Other Substances: colchicine		
	Respiratory Drugs: salmeterol		
	Urologic Drugs: avanafil, fesoterodine (in patients with moderate to severe renal or hepatic impairment), solifenacin (in patients with moderate to severe renal or hepatic impairment), darifenacin, vardenafil		
	Vasopressin Receptor Antagonists: conivaptan, tolvaptan		
	Analgesics: alfentanil, buprenorphine (IV and sublingual) oxycodone, sufentanil	Monitor for adverse reactions; dose reduction may be necessary	
	Antiarrhythmics: digoxin		
	Antibacterials: clarithromycin, trimetrexate, ciprofloxacin, erythromycin, clarithromycin		
	Anticoagulants and Antiplatelets: warfarin, cilostazol, dabigatran		
	Antidiabetic Drugs: repaglinide, saxagliptin		
	Antihelminthics, Antifungals and Antiprotozoals: artemether-lumefantrine, quinine		
	Antimigraine Drugs: eletriptan		
	Antineoplastics: bortezomib, brentuximab-vedotin, nintedanib, panobinostat, busulfan, ponatinib, erlotinib, ruxolitinib, gefitinib, sonidegib, idelalisib, tretinoin (oral), imatinib, vandetanib, ixabepilone		
	Antipsychotics, Anxiolytics and Hypnotics: alprazolam, aripiprazole, buspirone, cariprazine, diazepam, haloperidol, midazolam (IV), quetiapine, ramelteon, risperidone, suvorexant, zopiclone		
	Antivirals: daclatasvir, indinavir, maraviroc, cobicistat, elvitegravir (ritonavir-boosted), ombitasvir/paritaprevir/ritonavir with or without dasabuvir, ritonavir, saquinavir (unboosted), glecaprevir/pibrentasvir, tenofovir disoproxil fumarate		
	Beta Blockers: nadolol		
	Calcium Channel Blockers: diltiazem, other dihydropyridines, verapamil		
	Cardiovascular Drugsand Miscellaneous: bosentan, guanfacine		
	Contraceptives: dienogest, ulipristal		
	Gastrointestinal Drugs: aprepitant, loperamide, netupitant		
	Immunosuppressants: budesonide (inhalation), fluticasone (inhalation), budesonide (non-inhalation), fluticasone (nasal), methylprednisolone, ciclesonide (inhalation), tacrolimus (IV), tacrolimus (oral), cyclosporine (IV), cyclosporine (non-IV), dexamethasone		
	Lipid-Lowering Drugs: atorvastatin		
	Miscellaneous Drugs and Other Substances: alitretinoin (oral), cabergoline, cannabinoids, cinacalcet, galantamine, ivacaftor		
	SSRIs, Tricyclics and Related Antidepressants: venlafaxine		
	Urologic Drugs: dutasteride, oxybutynin, sildenafil (for erectile dysfunction), tadalafil (for erectile dysfunction and benign prostatic hyperplasia), tolterodine		
	Antineoplastics: regorafenib	Reduced concomitant drug concentration. Not recommended 2 weeks before, and during, itraconazole treatment	
	Gastrointestinal Drugs: *Saccharomyces boulardii*		
	Nonsteroidal Anti-Inflammatory Drugs: meloxicam	Reduced concomitant drug concentration. Concomitant drug dose increase may be necessary.	
	Antineoplastics: entrectinib, pemigatinib, talazoparib, glasdegib	Refer to the entrectinib, pemigatinib, talazoparib, and glasdegib prescribing information for dosing instructions if concomitant use cannot be avoided.	
	Antibacterials: isoniazid, rifampicin	Reduced itraconazole concentrations. Not recommended 2 weeks before, and during, itraconazole treatment	
	Anticonvulsants: carbamazepine, phenobarbital, phenytoin		
	Antivirals: efavirenz, nevirapine		
	Miscellaneous Drugs and Other Substances: lumacaftor/ivacaftor		
	Gastrointestinal Drugs: Drugs that lower gastric acid, such as antacids (e.g., aluminum hydroxide), H_2_ blockers, and proton pump inhibitors.	Reduced itraconazole concentrations. Use with caution. Administer acid neutralizing medicines at least 2 h before or 2 h after the intake of itraconazole.	
	Miscellaneous Drugs and Other Substances: valbenazine	Concomitant drug dose reduction is necessary. Refer to the valbenazine prescribing information for dosing instructions	
	Miscellaneous Drugs and Other Substances: eliglustat	For patients who are CYP2D6 extensive metabolizers (EMs) taking a strong or moderate CYP2D6 inhibitor, as well as for intermediate (IMs) or poor metabolizers (PMs), itraconazole is contraindicated during treatment and for two weeks after discontinuation. In CYP2D6 EMs not taking a CYP2D6 inhibitor, careful monitoring for adverse reactions is recommended, and a dose reduction of eliglustat may be required.	
Fluconazole	Antiarrhythmics: quinidine	Contraindicated	Fluconazole is a moderate inhibitor of CYP2C9 and CYP3A4, and a strong inhibitor of CYP2C19; co-administration with substrates of these enzymes may substantially decrease their clearance, increasing plasma concentrations and the risk of adverse effects, including hepatotoxicity, QT prolongation, and enhanced effects of other compounds metabolized by CYP2C19, CYP2C9, and CYP3A4.
	Antibacterials: erythromycin		
	Antipsychotics, Anxiolytics and Hypnotics: pimozide		
	Antihelminthics, Antifungals and Antiprotozoals: voriconazole	Avoid concomitant use. Should concurrent use be unavoidable, refer to concomitant drug-prescribing information for dosing instructions	
	Antineoplastics: olaparib		
	Antipsychotics, Anxiolytics and Hypnotics: lemborexant, lurasidone		
	Immunosuppressants: abrocitinib		
	Analgesics: alfentanil, methadone, fentanyl	Use with caution; monitor for adverse events. Dose adjustment may be necessary	
	Antiarrhythmic: amiodarone		
	Antibacterials: rifabutin, rifampin		
	Antidiabetic Drugs: tolbutamide, glyburide, and glipizide		
	Anticoagulants and Antiplatelets: warfarin, coumarin-type anticoagulants		
	Anticonvulsants: carbamazepine, phenytoin		
	Antineoplastics: cyclophosphamide, ibrutinib		
	Antipsychotics, Anxiolytics and Hypnotics: midazolam, triazolam		
	Antivirals: saquinavir, zidovudine		
	Calcium Channel Blockers: nifedipine, isradipine, amlodipine, verapamil, felodipine		
	Cardiovascular Drugs and Miscellaneous: losartan		
	Diuretics: tolvaptan		
	Immunosuppressants: cyclosporine, prednisone, sirolimus		
	Lipid-Lowering Drugs: atorvastatin, simvastatin, fluvastatin		
	Miscellaneous Drugs and Other Substances: vitamin A/retinoids		
	Nonsteroidal Anti-Inflammatory Drugs: celecoxib, naproxen, lornoxicam, meloxicam, diclofenac		
	SSRIs, Tricyclics and Related Antidepressants: amitriptyline, nortriptyline		
	Xanthine derivatives: theophylline		
	Antineoplastics: ibrutinib	Use with caution; monitor for adverse events. Refer to the ibrutinib prescribing information for dosing instructions	
	Antineoplastics: vinca alkaloids	May lead to neurotoxicity. Use with caution; monitor for adverse events.	
	Antipsychotics, Anxiolytics and Hypnotics: lurasidone	Avoid concomitant use. Should concurrent use be unavoidable, refer to lurasidone prescribing information for dosing instructions	
	Immunosuppressants: tacrolimus, tofacitinib	Monitor for adverse events; refer to the concomitant prescribing information for dosing instructions	
	Miscellaneous Drugs and Other Substances: ivacaftor and fixed dose ivacaftor combinations (e.g., tezacaftor/ivacaftor and ivacaftor/tezacaftor/elexacaftor)	Use with caution; monitor for adverse events. Refer to the ivacaftor (or ivacaftor combination) prescribing information for dosing instructions	
	Diuretic: hydrochlorothiazide	Increases fluconazole concentration	
	Antimicrobial: rifampin	Decreases fluconazole concentration	

Disclaimer: This table is intended for informational use only and may not apply uniformly in all regions. Healthcare professionals should refer to the latest product labels, regulatory recommendations, and local clinical guidelines to ensure appropriate prescribing within their jurisdiction.

## Data Availability

No new data were created or analyzed in this study. Data sharing is not applicable to this article.

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
