# Peer review of "Onychomycosis in Diabetics: A Common Infection with Potentially Serious Complications"

_life, 2025, doi:10.3390/life15081285_

Round 1

Reviewer 1 Report

Comments and Suggestions for Authors

1. Introduction
The authors are encouraged to provide more concrete examples illustrating the burden of onychomycosis among individuals with diabetes. For instance, including epidemiological data, complication rates, or case studies would strengthen the context. Additionally, the review would benefit from a clearly stated objective to help readers understand the scope of the paper, the key research gaps addressed, and the main unresolved issues in this field.

2. Section 2.2 – Pathophysiology and Clinical Presentation
The authors mention the role of non-dermatophyte molds (NDMs) in diabetic patients but should first present an overview of the most common causative agents of onychomycosis in the general population. This should include a list of frequently encountered dermatophytes, such as Trichophyton rubrum, which is considered the predominant pathogen. Following that, the discussion can transition into the occurrence of atypical or opportunistic pathogens in diabetic patients. This structure will improve clarity and contextual relevance for readers.

3. Figure 1
The biological representation of the fungal pathogen in Figure 1 needs revision. The fungus responsible for onychomycosis is filamentous and typically presents as hyphae/mycelium rather than oval or spherical forms, as shown in the current diagram. Furthermore, the diagram should accurately depict the pathogen's presence both on the surface and, more importantly, within the nail plate. A better understanding of fungal biology should be reflected in the figure's design.

4. Figure 2
Please clarify whether the photographs in Figure 2 are original or sourced from existing literature. If they are taken from other sources, they must be appropriately cited and labeled in accordance with copyright requirements.

5. Pathogen Identification and Diagnostic Methods
The review currently lacks a dedicated section that focuses on the pathogens responsible for onychomycosis, particularly regarding their identification and differentiation. This is a critical gap, as accurate diagnosis directly influences treatment decisions and outcomes. The authors should include information on diagnostic approaches (e.g., microscopy, culture, PCR, MALDI-TOF) and provide data on antifungal resistance, especially with regard to terbinafine and other commonly used agents.

6. Table 1
Table 1 should be expanded to include more detailed information, such as the specific pathogens identified in each case, the diagnostic method employed, and, importantly, the concentrations of antifungal agents used. This would greatly improve the table’s scientific and clinical utility.

7. Section 4.1 – Preventative Strategies and Preventing Relapse / Figure 4
It is recommended that the authors consider incorporating a diagnostic step into the prevention and management flow illustrated in Figure 4. Accurate diagnosis is crucial to the success of any subsequent therapeutic intervention.

8. Section 5 – Future Directions
The authors may consider discussing the current state of knowledge regarding antifungal susceptibility and resistance patterns among onychomycosis pathogens. Including this information would help define future research priorities and guide clinical practice.

Author Response

Reviewer 1 comments

  • Introduction: The authors are encouraged to provide more concrete examples illustrating the burden of onychomycosis among individuals with diabetes. For instance, including epidemiological data, complication rates, or case studies would strengthen the context. Additionally, the review would benefit from a clearly stated objective to help readers understand the scope of the paper, the key research gaps addressed, and the main unresolved issues in this field.

Authors: Thank you for your comment. We have added in a statement addressing the objective and the main points of the article (Page 2, Section 1, Lines 76-78). Additionally, we expanded on the global burden of onychomycosis, particularly in individuals with diabetes, and revised the third paragraph (Page 2, Section 1, Lines 56–75) to strengthen the context and relevance of the condition in this population.

  • Section 2.2: The authors mention the role of non-dermatophyte molds (NDMs) in diabetic patients but should first present an overview of the most common causative agents of onychomycosis in the general population. This should include a list of frequently encountered dermatophytes, such as Trichophyton rubrum, which is considered the predominant pathogen. Following that, the discussion can transition into the occurrence of atypical or opportunistic pathogens in diabetic patients. This structure will improve clarity and contextual relevance for readers.

Authors: Thank you for the helpful suggestion. We agree that including information on causative organisms in the general population helps contextualize the differences observed in diabetic patients. Accordingly, we have summarized common pathogens at the beginning of Section 2.2. (Page 3, Section 2.2, Lines 106-125).

  • Figure 1: The biological representation of the fungal pathogen in Figure 1 needs revision. The fungus responsible for onychomycosis is filamentous and typically presents as hyphae/mycelium rather than oval or spherical forms, as shown in the current diagram. Furthermore, the diagram should accurately depict the pathogen's presence both on the surface and, more importantly, within the nail plate. A better understanding of fungal biology should be reflected in the figure's design.

Authors: We appreciate the reviewer’s time and thoughtful critique, which have contributed meaningfully to the refinement of our work. After careful consideration, we have decided to remove the figure from the manuscript. Given the limited and evolving nature of the literature on these mechanisms, we felt it was best to avoid visual representation at this time.

  • Figure 2: Please clarify whether the photographs in Figure 2 are original or sourced from existing literature. If they are taken from other sources, they must be appropriately cited and labeled in accordance with copyright requirements.

Authors: Thank you for this important observation. We confirm that the photographs included in Figure 2 are original images captured by the authors specifically for the purposes of this manuscript. They have not been previously published or sourced from external literature.

  • Pathogen Identification and Diagnostic Methods: The review currently lacks a dedicated section that focuses on the pathogens responsible for onychomycosis, particularly regarding their identification and differentiation. This is a critical gap, as accurate diagnosis directly influences treatment decisions and outcomes. The authors should include information on diagnostic approaches (e.g., microscopy, culture, PCR, MALDI-TOF) and provide data on antifungal resistance, especially with regard to terbinafine and other commonly used agents.

Authors: Thank you for your insightful suggestion. We have added a new section (now Section 2.2) discussing diagnostic methods and common pathogens involved in onychomycosis, with emphasis on both the general population and diabetic patients (Page 3, Section 2.2, Lines 96-105). Additionally, we have expanded Section 5 to address antifungal resistance and the importance of antifungal stewardship (Page 5, Section 5, Lines 494-507).

  • Table 1: Table 1 should be expanded to include more detailed information, such as the specific pathogens identified in each case, the diagnostic method employed, and, importantly, the concentrations of antifungal agents used. This would greatly improve the table’s scientific and clinical utility.

Authors: Thank you for your comment. We would like to clarify that Table 1 is not a summary of individual clinical cases, but rather a synthesis of antifungal agents, their drug-drug interactions, and the underlying pharmacological mechanisms. As such, details such as specific pathogens, diagnostic methods, and antifungal concentrations are not applicable to this table’s content or purpose.

  • Section 4.1: It is recommended that the authors consider incorporating a diagnostic step into the prevention and management flow illustrated in Figure 4. Accurate diagnosis is crucial to the success of any subsequent therapeutic intervention.

Authors: Thank you for your valuable suggestion. While we agree that accurate diagnosis is critical to successful treatment, the intent of Figure 4 is to illustrate strategies aimed at minimizing relapse in diabetic patients who have already completed therapy. As such, diagnostic measures are not included, since they pertain to the initial disease presentation. Instead, Figure 3 highlights strategies to prevent onychomycosis and optimize therapeutic outcomes, including early and accurate diagnosis.

  • Section 5: The authors may consider discussing the current state of knowledge regarding antifungal susceptibility and resistance patterns among onychomycosis pathogens. Including this information would help define future research priorities and guide clinical practice.

Authors: Thank you for your comment. In response, we have included a paragraph addressing antifungal resistance and the role of antifungal susceptibility testing (AFST) in onychomycosis. We emphasize that prioritizing AFST and establishing surveillance systems for resistant dermatophytes may enhance antifungal stewardship and improve treatment outcomes in diabetic patients (Page 5, Section 5, Lines 496-507).

Reviewer 2 Report

Comments and Suggestions for Authors

Introduction 

The authors are advised to add a paragraph on onychomycosis in the current context and in relation to diabetes

The review abruptly starts with a heading and a subheading. I suggest authors signify the epidemiology part in the context of diabetes

Prevalence in the global context, including diabetes and co-morbidities, and other predisposing factors like ag,e can be tabulated

A separate section can be included: Etiology

A separate section on diagnosis may be useful to have a comprehensive discussion of the proposed topic

Treatment and other sections look OK

Author Response

Reviewer 2 comments

  • The authors are advised to add a paragraph on onychomycosis in the current context and in relation to diabetes.

Authors: Thank you for your valuable suggestion. We have briefly introduced onychomycosis in the Introduction section (Page 2, Section 1, Lines 67-75). For the purpose of this review, we have extensively discussed onychomycosis in the context of diabetes. The diagnosis of onychomycosis in the general context, as well as the spectrum of pathogens including dermatophytes and NDMs have been discussed further (Page 3, Section 2.2, Lines 96-125).

  • The review abruptly starts with a heading and a subheading. I suggest authors signify the epidemiology part in the context of diabetes.

Authors: We appreciate your insightful comment. In response, we have revised the subsection titles in Sections 2.1 and 2.3 to more clearly reflect the focus on individuals with diabetes.  

  • Prevalence in the global context, including diabetes and co-morbidities, and other predisposing factors like age can be tabulated.

Authors: Thank you for sharing your insights. We agree that onychomycosis is often associated with other comorbidities besides diabetes such as chronic venous disease, solid organ transplants, psoriasis and HIV; however, it is beyond the scope of this review to discuss other comorbidities as they require special considerations for management. We agree that age and diabetes are two interdependent factors that influences the prevalence of onychomycosis. Since we prepared a narrative review focused on issues in management, a systematic search of the literature was not performed to extract and tabulate numerical data such as age. At present, the inconsistent use of traditional and newer diagnostic methods also confounds prevalence figures; further studies utilizing PCR-based diagnosis are warranted to update the global prevalence of onychomycosis.

  • A separate section can be included: Etiology.

Authors: Thank you for your suggestion. Since the pathogen spectrum does not appear to differ significantly between diabetic and non-diabetic populations, we have opted to briefly describe the etiological agents (Page 3, Section 2.2, Lines 105-125).

  • A separate section on diagnosis may be useful to have a comprehensive discussion of the proposed topic.

Authors: Thank you for your suggestion. We agree that outlining diagnostic approaches and fungal identification is essential when discussing onychomycosis in diabetic patients. Accordingly, we have added Section 2.2 to address this (Page 3, Section 2.2, Lines 96-105).

  • Treatment and other sections look OK.

Authors: Thank you for your feedback. We appreciate you taking the time to review our work.

Reviewer 3 Report

Comments and Suggestions for Authors

There is definitely a need for a good review article on onychomycosis.  Here are some areas which should be upgraded in this manuscript:

1)  What is the expected length of treatment?

2)  When and to what extent should nail resection be performed?

3) How does onychomycosis relate to tinea pedis?

4) Is there a benefit to culture to identify potential resistance to treatment?

5)  Are there race differences in incidence of onychomycosis?

SPECIFIC COMMENTS

As the burden of diabetes continues to rise, particularly among older adults and so-56

cioeconomically disadvantaged populations, greater attention is being directed toward 57

secondary complications, such as onychomycosis, which is both underrecognized and 58

more commonly seen in underrepresented groups, such as low-income individuals, those 59

with less than a high school degree, Black and Hispanic persons [8]. Notably, diabetes is 60

the leading cause of lower limb amputation in the United States, most often following a 61

non-healing foot ulcer [9]. Diabetic patients are estimated to be up to three times more 62

likely to develop onychomycosis than non-diabetics, with reported prevalence rates rang-63

ing from 20% to 30% across multiple studies [10–14].Onychomycosis infection in diabetics 64

is linked to potential complications such as fungal and bacterial infections, and foot ulcer-65

ation [15,16].

YOU WANT TO TIE ONYCHOMYCOSIS TO FOOT ULCERS. MUST ELABORATE

Diabetic patients are at increased risk for onychomycosis due to a combination of 83

systemic, vascular, and behavioral factors. Established risk factors include older age, sex, 84

peripheral vascular disease, obesity, hypertriglyceridemia, and, poor glycemic control

HYPERTRIGLYCERIDEMIA? IS IT AN INDEPENDENT RISK FACTOR?

Author Response

Reviewer 3 comments

  • There is definitely a need for a good review article on onychomycosis. Here are some areas which should be upgraded in this manuscript:

1)  What is the expected length of treatment?

2)  When and to what extent should nail resection be performed?

3)  How does onychomycosis relate to tinea pedis?

4)  Is there a benefit to culture to identify potential resistance to treatment?

5)  Are there race differences in incidence of onychomycosis?.

Authors: Thank you for the insightful suggestions and for taking the time to read our article.

    • We have added the dosing regimen for each of the oral antifungals (Page 7, Section 3.2, Lines 230-231, 256-261, 278-280, and 287-289).
    • We have highlighted that surgical nail intervention is not recommended for patients with diabetes and should be reserved for patients when the benefits outweigh the risks (Page 2, Section 3.4, Lines 381-387)
    • We have added a paragraph to Section 2.3 discussing the close relationship between onychomycosis and tinea pedis, highlighting how tinea pedis may serve as a source of initial infection or contribute to reinfection in cases of onychomycosis (Page 3-4, Section 2.3, Line 136-143).
    • We have expanded on antifungal resistance and antifungal susceptibility testing in Section 5 (Page 5-6, Section 5, Lines 494-518)
    • Yes, Black and Hispanic individuals show a higher risk of onychomycosis (Page 5, Section 5, Lines 485-487).
  • YOU WANT TO TIE ONYCHOMYCOSIS TO FOOT ULCERS. MUST ELABORATE.

As the burden of diabetes continues to rise, particularly among older adults and socioeconomically disadvantaged populations, greater attention is being directed toward secondary complications, such as onychomycosis, which is both underrecognized and more commonly seen in underrepresented groups, such as low-income individuals, those with less than a high school degree, Black and Hispanic persons [8]. Notably, diabetes is the leading cause of lower limb amputation in the United States, most often following a non-healing foot ulcer [9]. Diabetic patients are estimated to be up to three times more likely to develop onychomycosis than non-diabetics, with reported prevalence rates ranging from 20% to 30% across multiple studies [10–14]. Onychomycosis infection in diabetics is linked to potential complications such as fungal and bacterial infections, and foot ulceration [15,16].

Authors: Thank you for your suggestion. We have revised lines 56-75 to more clearly convey the burden of onychomycosis in diabetic patients, including its potential role in the development of foot ulcers and risk of amputation (Page 2, Section 1, Lines 56-75).

  • HYPERTRIGLYCERIDEMIA? IS IT AN INDEPENDENT RISK FACTOR?

Diabetic patients are at increased risk for onychomycosis due to a combination of systemic, vascular, and behavioral factors. Established risk factors include older age, sex, peripheral vascular disease, obesity, hypertriglyceridemia, and, poor glycemic control 

Authors: Thank you for your insightful feedback and thorough review of our manuscript. We recognize that the inclusion of hypertriglyceridemia in that context may have introduced ambiguity. To improve clarity and precision, we have removed it from the sentence (Page 3, Section 2.3, Line 129). To our knowledge, hyperglycemia is not an independent factor for onychomycosis risk.

Reviewer 4 Report

Comments and Suggestions for Authors

The paper of Gupta and co-workers is well structured and has scientific sounding. However, there are some details that could be improved and information that could be added to further substantiate the problem presented and address some more recent strategies, particularly in terms of prevention and treatment.

Line 64 .”Onychomycosis infection in diabetics is linked to potential complications such as fungal and bacterial infections, and foot ulceration” The complication of onychomycosis is fungal and bacterial infections? Please review the sense of the phrase.

3.2. The authors wrote about the benefits of terbinafine. But there are some side effects that leave infected individuals to stop therapeutics. They could also refer those problems.

Line 361. The authors refer hygiene for preventing onychomycosis and the difficulty mainly for older people and men. But no reference to ways to obviate that were advanced. There are several papers describing how to avoid in-house transmission, examples of antiseptics, alternative azole formulations for feet application (nanosystems, liquid formulations), etc. A reference to one or more of these could increase the quality of the work.

Line 375. Authors refer textiles and the need for a proper decontamination. But the investigation in textiles is already producing textiles with silver, investigating the introduction of essential oils to prevent infections, and so on. It will be interesting if the authors could introduce this information in the paper.

Author Response

Reviewer 4 comments

  • The paper of Gupta and co-workers is well structured and has scientific sounding. However, there are some details that could be improved and information that could be added to further substantiate the problem presented and address some more recent strategies, particularly in terms of prevention and treatment.

Authors: Thank you for your time, thoughtful comments, and valuable suggestions to enhance the manuscript.

  • Line 64. “Onychomycosis infection in diabetics is linked to potential complications such as fungal and bacterial infections, and foot ulceration” The complication of onychomycosis is fungal and bacterial infections? Please review the sense of the phrase.

Authors: Thank you for carefully reviewing our work. We have revised the sentence to more clearly highlight the potential secondary complications associated with onychomycosis (Page 2, Section 1, Lines 62-64).

  • 2. The authors wrote about the benefits of terbinafine. But there are some side effects that leave infected individuals to stop therapeutics. They could also refer those problems.

Authors: We appreciate the reviewer’s suggestion and agree that this addition strengthens the manuscript. Accordingly, we have added a paragraph to Section 3.2 discussing terbinafine drug–drug interactions and the importance of monitoring for adverse events during concomitant use (Page 7-8, Section 3.2, Lines 242-251). 

  • Line 361. The authors refer hygiene for preventing onychomycosis and the difficulty mainly for older people and men. But no reference to ways to obviate that were advanced. There are several papers describing how to avoid in-house transmission, examples of antiseptics, alternative azole formulations for feet application (nanosystems, liquid formulations), etc. A reference to one or more of these could increase the quality of the work.

Authors: Thank you for your thoughtful comment. We have briefly addressed strategies to reduce transmission and improve hygiene at the individual level, including patient-centered education tailored to the specific needs, beliefs, and circumstances of diabetic patients (Page 3, Section 4.2, Lines 447-453).

  • Line 375. Authors refer textiles and the need for a proper decontamination. But the investigation in textiles is already producing textiles with silver, investigating the introduction of essential oils to prevent infections, and so on. It will be interesting if the authors could introduce this information in the paper.

Authors: Thank you for your thoughtful suggestion. Although we agree that there is a need for innovative strategies to prevent reinfection, we consider silver-embedded textiles and essential oils to be more investigational than practical for onychomycosis patients at this time. Further studies are warranted to validate the effectiveness of these interventions

Round 2

Reviewer 1 Report

Comments and Suggestions for Authors

The authors have done a commendable job in improving the manuscript. This review offers valuable insights into the issue of human onychomycosis.

Reviewer 4 Report

Comments and Suggestions for Authors

The authors review the manuscritp in accordance.